# Performance of LoRaWAN for Handling Telemetry and Alarm Messages in Industrial Applications

**DOI:** 10.3390/s20113061

**Published:** 2020-05-28

**Authors:** Francisco Helder C. dos Santos Filho, Plínio S. Dester, Elvis M. G. Stancanelli, Paulo Cardieri, Pedro H. J. Nardelli, Dick Carrillo, Hirley Alves

**Affiliations:** 1Federal University of Ceará, Quixadá 63902-580, Brazil; elvis.stancanelli@ufc.br; 2School of Electrical and Computer Engineering, University of Campinas, Campinas 13083-970, Brazil; pliniodester@gmail.com (P.S.D.); cardieri@decom.fee.unicamp.br (P.C.); 3Department of Electrical Engineering, Lappeenranta University of Technology, P.O. Box 20, FI-53851 Lappeenranta, Finland; pedro.nardelli@lut.fi (P.H.J.N.); dick.carrillo.melgarejo@lut.fi (D.C.); 4Centre for Wireless Communications, University of Oulu, P.O. Box 4500, 90014 Oulu, Finland; Hirley.Alves@oulu.fi

**Keywords:** LoRa, LoRaWAN, LPWAN, Industrial IoT

## Abstract

This paper analyzes the feasibility of the coexistence of telemetry and alarm messages employing Long-Range Wide-Area Network (LoRaWAN) technology in industrial environments. The regular telemetry messages come from periodic measurements from the majority of sensors while the alarm messages come from sensors whose transmissions are triggered by rarer (random) events that require highly reliable communication. To reach such a strict requirement, we propose here strategies of allocation of spreading factor, by treating alarm and regular (telemetry) messages differently. The potential of such allocation strategies has also been investigated under retransmission and diversity of gateways. Both indoor industrial plant and open-field scenarios are investigated. We compare the proposed solution with a benchmark scenario—where no alarm is considered—by using system level simulation. Our results show that it is possible to achieve high reliability with reasonably low delay for the alarm messages without significantly affecting the performance of the regular links.

## 1. Introduction

Digitization of processes is widespread in different aspects of our society, from residential energy management systems to scheduling systems of public services. This tendency is also happening in the industry (a group of related companies that produces goods or related services within an economy. For example, agriculture, forestry, fishing, mining, quarrying, oil and gas extraction, utilities, etc.). Concepts of Factories of the Future [1] or Industry 4.0 [2] have for already some time reached governmental bodies and engineering projects alike. Behind such concepts are the recent growth of Information and Communication Technologies (ICTs) and Industrial Cyber-Physical Systems (ICPSs) they will potentially create [3,4].

In particular, the upcoming fifth generation of mobile wireless networks (5G) and its standardization-related forums (e.g., 3GPP and 5G-PPP), guided by ICT industry, are putting a strong focus on the so-called “vertical applications.” These verticals are related to specific needs, systematized by the following classification (https://5g-ppp.eu/verticals/): Automotive, manufacturing, media, energy, eHealth, public safety, and smart cities. Within these verticals, three extreme operation modes have been identified in terms of end-application requirements [5], namely (i) massive connectivity, (ii) ultra-reliable and low latency communications (URLLC), and (iii) enhanced broadband. More recently, the discussions are moving beyond these three cases by identifying applications in which a combination of these modes is more suitable. For example, alarm messages may require high or ultra-reliability, but not extreme low latency required by feedback control in automation (in the order of milliseconds); a delay of seconds would be very acceptable for some alarm applications, leading to more alternatives to communication network design.

Besides, in most industrial applications, the traffic generated is of machine-type, which is significantly different from human-type [6]: machines tend to have periodic uplink traffic of relatively small data packets for telemetry, whereas humans have downlink dominated bursty traffic patterns with longer messages. Alongside the solutions available within Long-term Evolution (LTE) and 4G/5G, other technologies are currently offering a cheaper solution for long range coverage with low power devices (e.g., sensors). These solutions are [7,8]: Long-Range Wide-Area Network (LoRaWAN), SigFox, and Narrow Band-Internet of Things (NB-IoT) among others.

This paper focuses on the reliability and efficiency of regular and alarm messages with the goal of finding efficient ways of coordinating massive connectivity—the so-called massive machine-type communications (mMTC) [9], while allowing for reliable alarm messages for protection and security proposes in industrial environments. Therefore, we consider only applications that accept higher delays rather than “low latency” use-cases discussed by 3GPP. More specifically, we present here an analysis of a low-power wide-area network using LoRa [10]. Although some recent studies have been conducted to address the performance of LoRa in terms of its transmission range [11], energy efficiency [12], throughput [13] and reliability [14], none of them have provided a detailed study of how LoRa could be deployed in industrial settings where telemetry and alarms co-exist.

The present study proposes a simple yet effective method to differentiate alarm and regular messages based on the pseudo orthogonality of different spreading-factors of LoRaWAN. Our numerical results are based on ns-3 simulations and show that the proposed strategy in combination with a limited number of retransmissions leads to full reliability in most cases, with relatively small latency, which is bounded by a function of the number of retransmissions.

The rest of this paper is divided as follows. Section 2 provides a thorough review of other relevant works in Low Power Wide Area Network (LPWAN) applied in industrial settings, evincing our contributions in relation to the current state-of-the-art. Section 3 introduces the technical details of LoRa and LoRaWAN. Section 4 presents the system model used here. In Section 5 we present our numerical results and the discussions about the potential of LoRaWAN in industrial environments. Section 6 concludes this paper.

## 2. Related Works and Our Contributions

### 2.1. Literature Review

Currently, there is a growing interest on LPWAN technology like LoRa and IEEE 802.11ah [15,16,17], in which some focus is given on the use of LoRaWAN in Industry 4.0 applications and other studies focus on alarm events triggered by protection systems [18,19,20]. For instance, in [15], the authors investigate LoRa technology for the implementation of industrial wireless networks suitable for sensors and actuators in an industrial setting and discuss the viability of using LoRaWAN technology as compared to traditional industrial wireless systems. Another example comes from the smart city scenario, which is discussed in [17]. Authors demonstrate via ns-3 based simulations that LoRaWAN network provides higher throughput than other technologies that resort to pure-ALOHA access models.

Already in [16] an assessment is made for typical Industrial Internet of Things (IIoT) on indoor industrial monitoring applications. The authors propose a comparison of the LoRaWAN network with the IEEE 802.15.4 network protocol, in which the results confirm that LoRaWAN can be considered as a viable technology for IIoT. In [18] a study of the reliability and efficiency for regular and alarm machine-to-machine traffic under an 802.11ah network was conducted, in which an access mechanism tailored for telemetry is proposed. The authors of [19] carried out an overview and study of the industrial alarm systems, contributing to the major causes of alarm overload in many industrial alarm systems, which play critically important roles for the safe and efficient operation of modern industrial plants. An investigation of the feasibility of alarms for a smart metering scenario is done in [20], employing simulation and mathematical modeling.

### 2.2. Our Contributions

The recent works listed before have confirmed the viability of LoRaWAN for industrial applications. In this work, we focus on such technology to analyze three different benchmark scenarios illustrating the capabilities of LoRaWAN using a system level simulator ns-3. In particular, we study in this paper the following industrial settings: (i) indoor industrial plant [21]; (ii) open-field with one gateway industrial environment, both with one gateway and retransmissions only for alarms; and (iii) open-field with four gateways in an industrial environment without retransmissions for alarms (e.g., mining, power lines, agriculture fields).

We have compared different ways of handling the alarm messages and our results show that it is indeed possible to achieve high reliability in alarm messages without affecting the performance of the regular and telemetry communication links. We propose an allocation scheme that assigns a different spreading factor (SF) to the device depending on whether the event is regular or alarm, while alarms are also allowed to have a limited number of retransmissions. To the best of our knowledge, this is the first time that such an approach is proposed for LoRaWAN. In this sense, the analysis to be presented here indicates the potential of using LoRa to handle (random, less frequent) alarm messages while it is capable of delivering them with high reliability and causing a negligible performance loss in the regular messages.

## 3. LoRa and LoRaWAN Overview

LoRaWAN is a LPWAN protocol. It defines medium access control (MAC) features and message formats that are delivered to a proprietary physical layer called long range (LoRa). LoRaWAN is developed and continually updated by the LoRa Alliance and LoRa was developed by Semtech Corporation. LoRaWAN networks are typically deployed in a star-to-stars topology in which gateways relay messages between end-nodes and a central network server (NS), and NS routes the packets from each device to the associated application server. Gateways and the network server are connected using secured standard internet protocol (IP) connectivity. On the other side, end-nodes are connected to one or many gateways using single-hop LoRa [22]. More specific details of LoRa and LoRaWAN are described in the following subsection.

### 3.1. LoRa

LoRa is a proprietary physical (PHY) layer derived from chirp spread spectrum (CSS) technology. The nominal bit rate ranges from 0.3 kbps to 27 kbps. For a modulation bandwidth (BW), the rate $R_{b}$ is a function of the spreading factor (SF) employed, and is given by [10]:(1)$$R_{b}=\mathrm{SF}\;\frac{\left[\frac{4}{4+{\mathrm{CR}}_{i}}\right]}{\left[\frac{2^{\mathrm{SF}}}{\mathrm{BW}}\right]}$$
in which SF can assume integer values between 7 and 12, and CR${}_{i}$ is the code rate index that can take integer values between 1 and 4 [23], and the coding rate is fixed at 4/5 or 4/6 for the LoRaWAN protocol.

A variety of bandwidths are available, some of them are [24]: 125 kHz, 250 kHz, and 500 kHz. Furthermore, transmissions with different spreading factors are somewhat quasi-orthogonal [25,26] to each other, increasing network capacity. In general, for projects with strict reliability requirements (without focusing in data rate) the most suitable option for the possible technical settings is to maximize the coding rate index to 4 while boosting link budget to minimize bandwidth and maximize spreading factor (SF=12).

The modeling of the PHY layer contains two key factors of LoRa, namely sensitivity and quasi-orthogonality [25,26], to decide whether a transmission is successfully received. The sensitivity thresholds were obtained from the Semtech’s datasheet [23], as shown in Table 1 (values in dBm).

A Signal-to-Interference Ratio (SIR) value is calculated for each signal that is interfering in the same channel of the desired signal (please note that we do not assume inter-channel interference). This value is then compared to a threshold reported in [27] as presented in Table 2. The SIR values are given in dB, where the rows refer to the SF of the desired signal, whereas the columns refer to the interfering signal’s SF that is currently being considered. If the SIR calculation is above the tabulated threshold of Table 2, the packet is successfully received and forwarded to MAC layer.

The MAC layer creates a series of objects to keep track of available transmission time and limit transmission since LoRaWAN operates in an unlicensed band so subject to duty cycle restrictions.

In [28], LoRaWAN regional parameters are specified listing the unlicensed industrial, scientific, and medical (ISM) bands for the different regions around the world. For example, in Europe, LoRa operates in 868 MHz, which is regulated by the norm UE868MHz. In this case, the maximum transmission power should be below 14 dBm and maximum duty cycles should be 1% for sub-bands g1 (868.0–868.6 MHz) as regulated by ETSI EN300.220 recommendation.

### 3.2. LoRaWAN Networks

LoRa system architecture is used by LoRaWAN networks to support two important requirements as battery lifetime and long-range connectivity. A LoRaWAN network consists of one or more LoRaWAN gateways that are all connected to one central network coordinator, or so called NS. This architecture is depicted in Figure 1.

LoRaWAN gateways are basic protocol bridges. Each gateway receives LoRa modulated radio messages from all LoRaWAN end-nodes. Every received LoRaWAN frame with a correct cyclic redundancy check (CRC) code will be forwarded to the NS encapsulated in an IP frame. One strategy to prolong LoRaWAN end-nodes battery life is increasing the number of gateways in a given area since the distance between end-nodes and gateways would decrease.

The LoRaWAN defines end-nodes like class A, class B or class C. Class A supports bidirectional communication, the uplink message being mandatory, where the device can send an uplink message at any time and in the sequence opens two reception windows, used by NS to confirm message, at specified times (1 s and 2 s) respectively. Class B differs from class A by adding scheduling of the receive window for downlink message from the network server, and class C differs from class A by keeping the receive window open unless they are transmitting. In this architecture (Figure 1), the NS manages the LoRaWAN network.

## 4. System Model and Proposed Solution

The proposed analyses of LoRa are conducted with a focus on the up-link communication so that both alarm and telemetry messages are transmitted from end-nodes to the gateways.

The system simulations are executed regarding the EU863-870 ISM band with bandwidth of 125 kHz, and coding rate 4/5 [28]. The multiple channels, which are established throughout the cell, follow models composed mainly of path-loss (outdoor scenario) and including obstacles (indoor scenario) that affect the propagation profile. In these cases, buildings are uniformly created and regularly placed along a two-dimensional grid.

The positions of the gateways are predetermined. All end-nodes are class *A* devices (refer to [29]) and their positions are differently established depending on if they are alarm nodes or regular nodes. The regular nodes are randomly placed (uniformly distributed along the cell area), but they are kept fixed during the simulation running time, and each node generates 28-bytes packets, periodically transmitting at intervals of 600 s, and the first transmission of each node is decided by random delay via a uniform random variable, i.e., ∼U([0,600]). The alarm nodes are also kept fixed during the whole simulation running time, but they are spatially distributed in three different ways: (i) random, uniformly distributed over the two-dimensional grid; (ii) deterministic, distributed as a star topology; or (iii) at random, uniformly distributed along with the angles over a ring with a predetermined radius and centered in the simulation area (corresponding to a single cell). These alarms generate 14-bytes packets, and randomly transmit at the time defined by the exponential distribution of mean 600 s, i.e., ∼$\mathrm{Exp}(\frac{1}{600})$.

We investigate the system performance considering one or four gateways in a single cell comprising hundreds or a few thousands of regular nodes, but no more than 20 alarm nodes [18,19]. The LoRa simulations deal with PHY and MAC layers, whereas the LoRaWAN extends to the gateways. The traffic from gateway toward network server is assumed ideal without any external interfering source.

The spreading factor for each regular node is chosen and allocated as the lowest one providing adequate receiving sensitivity (estimated based on reception power compared to Table 1) using transmission power of 14 dBm [28]. This strategy is henceforth called SF basic.

In cases with four gateways, the aforementioned estimate is obtained for the gateway receiving the strongest version of the signal transmitted from the regular node. When dealing with alarm nodes, we can still follow the same basic strategy or we can use another one that best fits its critical characteristics, providing some sort of priority to these devices. Here, we will study two specific strategies, namely SF shift and SF reservation. They are expected to provide an improved robustness to the alarm messages.

In the SF shift strategy, each alarm node is allocated to the SF immediately higher than the one that would be obtained from basic strategy, respecting the maximum SF available (say SF12). For example, if an alarm node would be allocated as SF9 in the basic strategy, by following the SF shift strategy, it will become SF10. This, however, does not avoid that the regular and alarm nodes share the same SF, potentially interfering to each other.

In the SF reservation, we apply the SF basic strategy for all alarm nodes as a first step, except that no allocation is hitherto done. Based on all SF values estimated as adequate for the alarm nodes, we choose the highest one, and adopt it as the reserved SF (rSF). Then, all the alarm nodes will be allocated to this rSF regardless the estimates of the first step, providing an improved robustness, though at the expense of reduced throughput. At the same time, the regular nodes must not be allocated to rSF. Rather than regular node being allocated to this rSF, they will be allocated to the immediately neighboring SF, preferably the higher (rSF + 1); the only case that rSF + 1 cannot be used to replace rSF for regular node is when rSF is the maximum SF available and, therefore rSF + 1 is not available. All alarm nodes must be allocated to a single and reserved SF, which is rSF, whereas regular nodes can be allocated to any SF except rSF. As a first example, if rSF is SF10, any regular node that would be allocated to SF10 will be allocated to SF11 if it is available. As a second example, if rSF is SF11, any regular node that would be allocated to SF11 will be allocated to SF12 if it is available. As a final example, if rSF is SF12 and there is no higher SF available, any regular node that would be allocated to SF12 will remain allocated in SF12. Although this strategy is not a good choice, it was considered in the work on basis of comparison with other strategies.

Other techniques can also be employed to improve the reliability of the transmissions. So, temporal and spatial diversity techniques are exploited separately. In the first case, replicas of alarm information are sent whenever ACK is not received within the two reception windows. This retransmission procedure is repeated until a maximum number is reached; such a number has to be predetermined for each scenario. The reliability of the alarm messages is then improved. Note that retransmissions are not allowed for regular nodes just to not increase the traffic offered in the network.

Already for the case of spatial diversity, both alarm and regular messages are benefited, by using more than one gateway. In our simulations, we consider one case with four gateways (to be described in more details later); this number was selected so that all devices can find at least one gateway that allows them to use SF7, as illustrated in Figure 2c. As higher the number of gateways, better will be the spatial diversity; it is enough that any of the gateways succeeds in receiving the message.

## 5. Numerical Results and Discussion

In this section, we investigate the performance of LoRa considering the coexistence of telemetry and alarm messages as previously discussed. We rely on numerical results obtained from computational simulations using ns-3 (At this point, it is unfeasible to study the proposed scenario in a test-bed since (i) it requires a large number of nodes distributed in a large region and (ii) it is not possible to change the SF allocation policy due to proprietary policies of LoRa. In fact, one of our goals is to show that it is worth having a smart treatment of different types of messages.). We carried out this study considering three generic but illustrative scenarios: indoor industrial plant (IIP), open field one gateway (OF1GW), and open field four gateways (OF4GW).

In all three, the benchmark case assumed the scenario without alarm messages. When alarms were considered, a limited number of retransmissions were allowed to improve the packet delivery reliability in the IIP and OF1GW cases. In the case of OF4GW, the spatial diversity technique in the gateways was used (instead of retransmission of alarms) to achieve high reliability. They were treated in three different ways (as previously presented): (i) as a regular message with no priority, (ii) as a priority message according to SF shift strategy, or (iii) as a priority message according to SF reservation strategy.

Different topologies of the alarm nodes were considered, namely alarm topologies: (a) Uniform (random), (b) “Star edges” (deterministic), and (c) Orbital (random) with regular distances in 1D, but random angle. The combinations of these topologies and SF allocations for the alarms formed the configurations that are gathered in Table 3. For simplicity of notation, these configurations were coded as [Alarm topology id] - [SF allocation id]. For example, star topology with SF shift was referred as to “b-ii”. For all cases, we considered only one or four gateways (although we could consider more, especially in the case of the open field) and the simulation time corresponded to 12 h.

A snapshot relating positions and SFs of the devices is shown in Figure 2 for each investigated scenario. Note that as far as the device (either regular or alarm node) was from the gateways, a higher SF was required. The specifications of these three scenarios are given in the following. (The codes used in this paper are available at https://github.com/signetlabdei/lorawan.)

### 5.1. Indoor Industrial Plant

This scenario had the following characteristics:From 100 to 500 end-nodes, among which 10 alarms;The coverage area is a circle with radius of 500 m;Retransmission for alarms, maximum of 8 attempts;One gateway in the center of the circle;The channel considers obstacles.

For this scenario, as for all others, the first metric investigated was the total throughput comprising all regular nodes. This throughput was measured by counting the packets transmitted successfully, in terms of bits, and then dividing by the simulation time in seconds. Let us define a simulation campaign as an extensive and complete simulation process, from the running start until its closure, together obtaining the results. The execution of a few independent campaigns proved to be sufficient to reveal reliable results. Five simulation campaigns were run for each one of the nine configurations listed in Table 3 as well as for the benchmark case; the obtained results of the regular nodes throughput versus the number of end-nodes (which comprises both regular and alarm nodes) increase linearly at 37 kbps every 100 nodes and the largest margin of error obtained (among all evaluated loads) was of 1.61 bps for a 95% confidence level.

Although the scenario included obstacles, the coverage area was small (in relation to LoRaWAN capabilities) and, therefore, the small numbers of alarm nodes with their rarer messages could not degrade the regular communication performance. The different impacts between the SF reservation and SF shift strategies come from the fact that the regular nodes were disadvantaged because they could not use the SF that was reserved by alarms.

The second investigated metric was the packet success rate of transmission, which was computed as the ratio of the number of successfully transmitted packets over the total number of packets. The packet success rate of transmission relative to regular and alarm nodes is presented as a function of total number of end-nodes in Figure 3. The 95% confidence level margins of error were computed for all evaluated loads, but only the largest value was presented per curve.

This figure shows a visible change only for SF reservation strategy, unlike the SF shift strategy that did not change the reliability of the regular nodes. The reservation of SF in favor of alarm nodes restricted the resources available to the regular nodes, which may have crowd them into another inappropriate SF. We also present in Figure 3 the probability of successful of all alarm nodes with retransmission, which achieved 100% success in all scenarios (plotted on the line with lilac stroke).

Already in Figure 4 the average delay is presented (*y*-axis plotted in log scale), where we had the worst case for SF reservation getting closer to 80 ms, being a very low delay compared to other cases, but that greatly impaired the performance of regulars. For the SF basic and SF shift cases, the average delay was around 100 ms, keeping the reliability of the regulars above 98%, and it is worth mentioning that the strategy of SF shift was only feasible for a small number of alarms; a large number of alarms led to a long time-on-air, increasing the probability of collisions as presented in [20].

### 5.2. Open Field with One Gateway

Another five simulation campaigns were obtained, regarding the open field scenario characterized by:from 500 to 2000 end-nodes, among which 1% alarms.the coverage area is a circle with radius of 6 km;retransmission for alarms, maximum of 8 attempts;one gateway in the center of the circle;the channel has no obstacles.

Now the largest margin of error obtained was of 34.15 bps for a 95% confidence level. In this case, it is worth mentioning that there was a greater diversity in the SF allocations due to the gateway coverage. However, such diversity was solely based on the distances since there were not obstacles.

By looking at the packet success rate of regular nodes transmissions, in Figure 5, we note that the performance is not consistently high. As in the IIP the reliability for SF reservation strategy is visibly different from the other cases, except for the no-alarm case that showed the best performance ever. It shows that the performance of the regular messages is greatly impaired in order to keep the alarm messages with high reliability, as seen in Figure 5.

The average delay is shown in Figure 6, in which the smallest average delay was 275 ms, which was 10 ms less than the worst case of the IIP, this happened because of the greater number of retransmissions that was required to achieve higher reliability for the alarm messages. In other words, the “single-shot” reliability of this scenario was lower, and then more retransmissions were needed at expense of higher delays.

### 5.3. Open Field with Four Gateways

Respective to the previous scenario, four gateways were regularly distributed over the coverage area. This new scenario was characterized by:from 500 to 2000 end-nodes, among which 1% alarms.the coverage area is a circle with radius of 6 km;four gateways in a co-centered ring with radius of 3.5 km;no retransmissions;the channel has no obstacles.

Another five simulation campaigns were obtained such that the largest margin of error for regular nodes throughput was of 2.37 bps for a 95% confidence level. Similar observations of the previous scenarios could be reproduced here, both for throughput and packet success rate metrics, however, with respect to the differences of absolute values. At the maximal load (2000 end-nodes), the benchmark case’s throughput with four gateways was 16.60% greater than with one gateway, with values increasing linearly by 37 bps every 100 nodes.

In terms of the probability of success of regular node transmissions, this proportion corresponded to an increase of 16 percentage points in favor of the use of four gateways, while for the probability of success of alarms we had a value of 98.45% in the worst case, as seen in Figure 7. As this scenario did not assume retransmissions, the delay came from the time-on-air, therefore we will not present it here.

### 5.4. Discussion

Before obtaining the results achieved, it was expected that the SF allocation strategies and the size of the network would affect performance, but it was not known how. The results are representative to demonstrate that the proposed idea can lead to improved performance. However, this does not imply that the proposed solution is always the most appropriate. The study of this work indicates that, for industrial environments, treating alarms differently is an interesting option. However, the actual implementation must be evaluated on a case-by-case basis, and the main contribution of this work is to prove that the construction of different strategies of the LoRa network to deal with alarms can bring benefits to the network performance.

The results presented in the previous section answer positively the initial question posed by this paper: Is it possible to handle regular telemetry and alarm messages in industrial environments using LoRa?

However, we also showed that there is not an optimal strategy to handle alarm messages in terms of their reliability considering the three proposed representative scenarios. This indicates that, while it is indeed possible to achieve 100% of packet success rate for the alarm messages without perceivable impact in the regular messages performance, there is still a strong scenario-dependence in relation to the best approach to manage alarm messages. This would require an analysis of the designing trade-offs involved as well as a sensitivity study for these variables.

The results presented in previous subsections show that scenarios with retransmissions represent the best choice for industrial applications that can accept some delay. We showed that it is possible to achieve 100% reliability, reaching the ultra-reliable regime, with a bounded delay (but not extremely low as required for feedback control) in all studied cases for the alarms without negatively affecting the regular transmissions. This strategy is of lower cost in comparison with the deployment of spatial diversity via new gateways. Retransmissions would be then the best option when latency is not a major requirement for specific industrial applications that the communication system based on LoRaWAN needs to be deployed. On the other hand, the SF reservation proved to be a non-viable option in all scenarios as it greatly impairs the performance of regulars.

Our results reinforce the need for using system level simulations before the actual deployment of LoRa where alarm and telemetry nodes co-exist. Such an approach allows for testing different deployment options and the technological feasibility for the specific industrial setting under consideration. In practical terms, the communication system engineering team, for instance, would need to characterize:the industrial area and coverage based on the actual gateway position;the way that the telemetry sensors would be positioned as well as their messages length and periodicity;the structure (i.e., what is the event that the alarm is related to) and length of the alarm messages;possible sensor mobility;channel characterization: line-of-sight or not, existence of barriers (machines, walls), sources of multi-path.

With these and other characteristics in hand, the ns-3 simulation as proposed here can be employed to test different setups for the alarm to evaluate the best performing alarm-system design. This is possible including real maps of industrial plants.

## 6. Conclusions and Future Research Directions

This paper investigated the possibility of using LoRa as a communication network in industrial scenarios where regular and alarm sensors co-exist. Our results consistently showed that high reliability can be obtained for alarm nodes without negatively affecting the regular nodes’ throughput. However, we could infer from our results that the SF reservation is not a good solution because it blocks the use of SF for the regular nodes. The SF shift can be a good strategy for low numbers of end-nodes, or in cases of gateway spatial diversity [13] that assigns most of the end-nodes with low SF. We tested nine possibilities of SF allocation and alarm topologies; all of them generally provided good performance.

This opens the possibilities to further studies looking at either specific industrial environments (e.g., wind farms, a car manufacture, or monitoring of power lines), or more theoretical studies based on stochastic geometry and queuing theory. Other future studies shall consider a detailed sensitivity analysis of the designing parameters (e.g., relative size of telemetry and alarm messages, or frequency of regular messages), optimization of the trade-offs regulated by such a parameters and adaptive “on-line” strategies to set them based on the actual situation of the network. For that, we intend to develop a mathematical model of the packet success probability, to serve as a guide in the optimization. A first approach would be to model the alarm packet arrival process as a Poisson point process. In this scenario, we will also focus on the impact of the number of alarm nodes in the communication system performance. In addition, we expect to carry out small scale field experiments to validate the results obtained with the mathematical model and the simulations in ns-3.

## Figures and Tables

**Figure 1 sensors-20-03061-f001:**
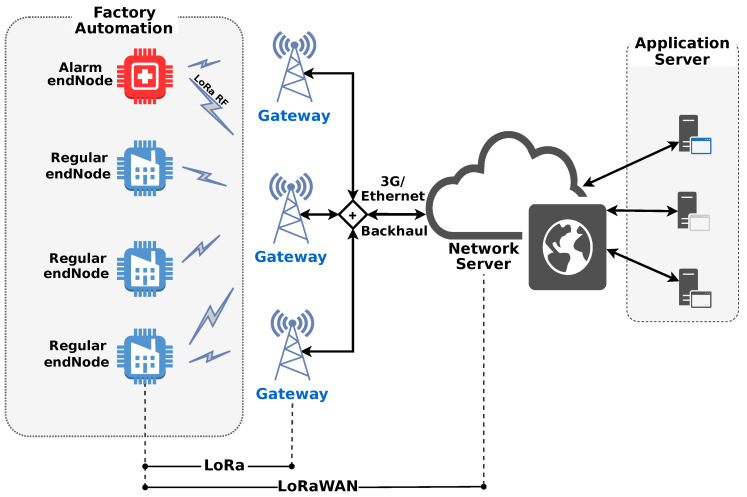
LoRaWAN architecture exemplified for industrial applications. The main elements are: end-nodes illustrated by sensors, gateways (depicted as base stations), network server (NS).

**Figure 2 sensors-20-03061-f002:**
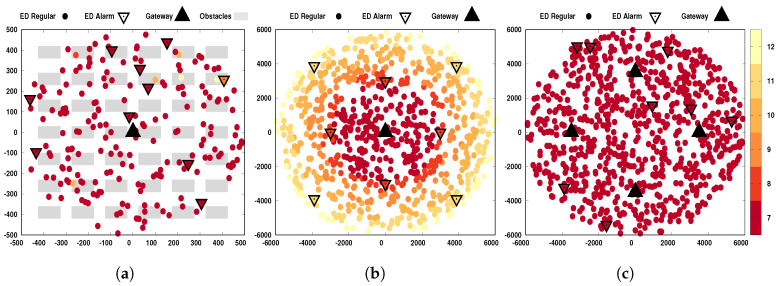
Sample of bidimensional spatial distribution of SF allocation for the three scenarios: (**a**) at the left-hand side, indoor with one gateway, 190 regular and 10 alarm nodes; (**b**) at the center, open field with one gateway, 792 regular and eight alarm nodes; and (**c**) at the right-hand side, open field with four gateways, 792 regular and eight alarm nodes. The marker color of each end-node (ED) changes according to the SF allocated to it. Abscissa and ordinate axes refer to distances in meters.

**Figure 3 sensors-20-03061-f003:**
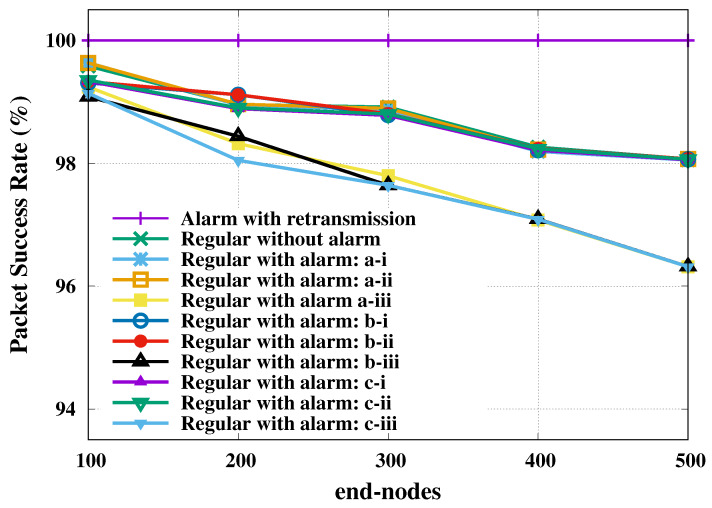
Regular and alarm nodes’ packet success rate for indoor industrial plant. Regarding a 95% confidence level, the largest margins of error, in percentage points, are: 0.000 (alarm with retransmission); 1.186 (regular without alarm); 0.904 (regular with alarm: a-i); 0.905 (regular with alarm: a-ii); 1.106 (regular with alarm: a-iii); 0.668 (regular with alarm: b-i); 0.663 (regular with alarm: b-ii); 0.927 (regular with alarm: b-iii); 0.777 (regular with alarm: c-i); 0.765 (regular with alarm: c-ii); and 1.390 (regular with alarm: c-iii).

**Figure 4 sensors-20-03061-f004:**
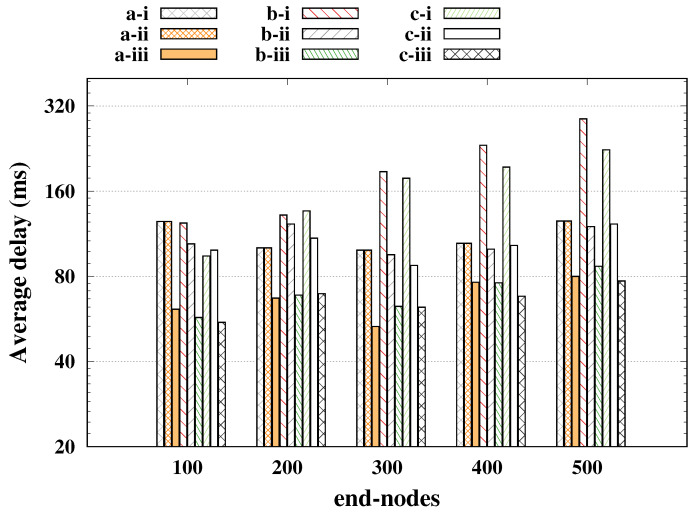
Alarm nodes’ delay for indoor industrial plant.

**Figure 5 sensors-20-03061-f005:**
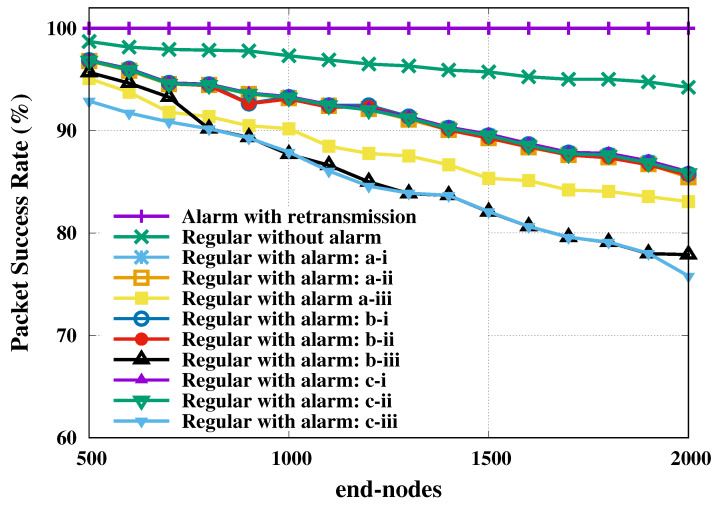
Regular and alarm nodes’ packet success rate for open field and one gateway. Regarding a 95% confidence level, the largest margins of error, in percentage points, are: 0.034 (alarm with retransmission); 1.297 (regular without alarm); 1.297 (regular with alarm: a-i); 1.276 (regular with alarm: a-ii); 5.560 (regular with alarm: a-iii); 1.264 (regular with alarm: b-i); 1.272 (regular with alarm: b-ii); 2.053 (regular with alarm: b-iii); 1.301 (regular with alarm: c-i); 1.293 (regular with alarm: c-ii); and 2.231 (regular with alarm: c-iii).

**Figure 6 sensors-20-03061-f006:**
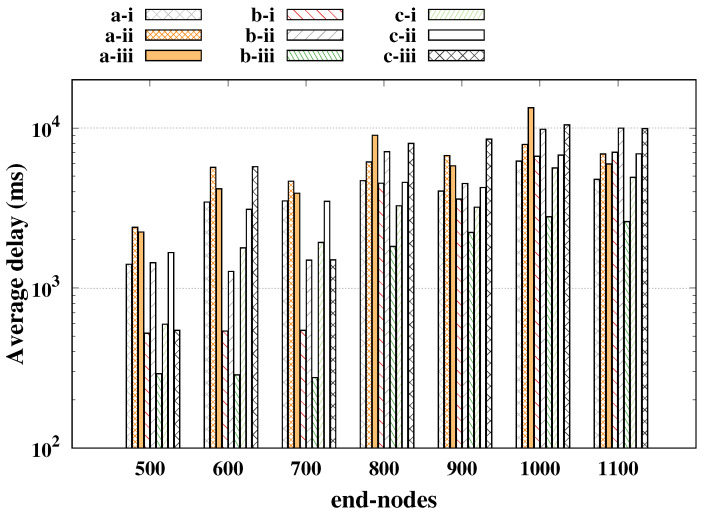
Alarm nodes’ delay for open field and one gateway.

**Figure 7 sensors-20-03061-f007:**
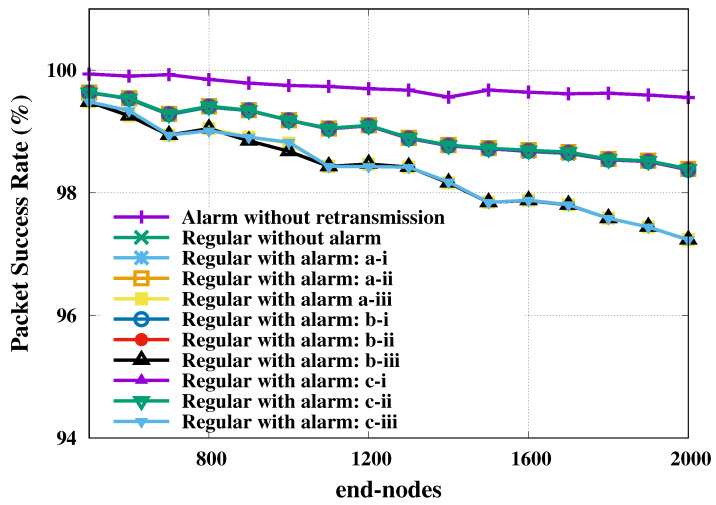
Regular and alarm nodes’ packet success rate for open field and four gateways. Regarding a 95% confidence level, the largest margins of error, in percentage points, are: 0.177 (alarm without retransmission); 0.300 (regular without alarm); 0.299 (regular with alarm: a-i); 0.299 (regular with alarm: a-ii); 0.490 (regular with alarm: a-iii); 0.304 (regular with alarm: b-i); 0.299 (regular with alarm: b-ii); 0.422 (regular with alarm: b-iii); 0.297 (regular with alarm: c-i); 0.299 (regular with alarm: c-ii); and 0.345 (regular with alarm: c-iii).

**Table 1 sensors-20-03061-t001:** The sensitivity threshold (in dBm) for a specific spreading factor (SF).

SF7	SF8	SF9	SF10	SF11	SF12
−124	−127	−130	−133	−135	−137

**Table 2 sensors-20-03061-t002:** Thresholds of Signal-to-Interference Ratio (SIR) (values in dB) for all spreading factor combinations of the desired and interfering signals [27].

Desired Signal	Interfering Signal					
	SF7	SF8	SF9	SF10	SF11	SF12
SF7	6	−16	−18	−19	−19	−20
SF8	−24	6	−20	−22	−22	−22
SF9	−27	−27	6	−23	−25	−25
SF10	−30	−30	−30	6	−26	−28
SF11	−33	−33	−33	−33	6	−29
SF12	−36	−36	−36	−36	−36	6

**Table 3 sensors-20-03061-t003:** Configurations of alarm topology and SF allocation.

SF Allocation	Alarm Topology		
	a	b	c
**i**	distributed uniformly and SF basic	distributed in star and SF basic	distributed in orbits and SF basic
**ii**	distributed uniformly and SF shift	distributed in star and SF shift	distributed in orbits and SF shift
**iii**	distributed uniformly and SF reservation	distributed in star and SF reservation	distributed in orbits and SF reservation

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
