# Peer review of "Performance of LoRaWAN for Handling Telemetry and Alarm Messages in Industrial Applications"

_sensors, 2020, doi:10.3390/s20113061_

Round 1

Reviewer 1 Report

The topic is interesting for Industry 4.0 development, as the manuscript provides a guide to select among different configurations of LoRaWAN deployments, which can be useful for network designers.

I miss some experiments in real-world environments that would support the simulation results, confirm the validity of the conclusions (actual environments introduce unpredictable elements that are not considered by simulators), and improve the scientific value of the contribution.

I also suggest to modify the design of figure 2, enlarging the three plots in order to improve the readability. Besides, please check the readability of the other figures when printed in black and white printers.

Please, check the references: i.e. [1] is not complete.

Please, review the text: there are some typos and repeated words.

Reviewer 2 Report

- The reference [6] is quite old; a fresher one is recommended
- In paragraph 3 expand acronyms before use (e.g. NS, CSS)
- "the coding rate index to 4" means that the system is no more LoRaWAN since LoRaWAN use coding rate 4/5 or 4/6
- Table 2 is not correct as demonstrated by reference 25
- The duty cycle is not regulated by ETSI EN 300220 but by CEPT ERC 70-03 recommendation
- In Fig.2 alarm nodes are hardly distinguishable and the entire figure is unreadable
- In the caption of Fig.2 "b-i configuration" is unclear
- The SF reservation strategy is going to "kill" all the regular nodes needing SF12 is the rSF=12
- Why considering the SF reservation strategy if "the SF reservation proved to be a non-viable option in all scenarios as it greatly impairs the performance of regulars"? A paper is not supposed to report all the failed attempts to reach a conclusion, except if anybody has previously proposed a solution and the paper demonstrates it does not work; the SF reservation strategy is clearly not working without the need for any simulation and nobody has proposed it previously.
- The title of the paper is "Performance of LoRaWAN for Handling Telemetry
and Alarm Messages in *Industrial* Applications": I don't see any reason to consider open field scenarios
- 100% success probability is highly difficult to estimate: how many packets where sent?

Reviewer 3 Report

In this paper, the authors focused on LoRaWAN for industrial applications and analyzed three different benchmark scenarios to illustrate LoRaWAN capabilities. They proposed "strategies of allocation of spreading factor, by treating alarm and regular (telemetry) messages differently".

I liked the paper but I have some comments.

Suggestions and questions (answers can be used to improve the paper):
1- Consider the text "Five simulation campaigns were run for each one of the nine configurations..."
1.1- What do the authors call "campaign"? Is it repetition? It is unclear.
1.2- Why exactly five? Could this number be greater?
1.3- Were outliers discarded?
2- Discussion section must be improved:
2.1- What results were expected (or predictable) before simulations? were there unexpected results?
2.2- What are the limitations of the study? The authors should acknowledge limitations, including the generalization of the results, representativeness of the scenarios, etc.
2.3- Moreover, the relevance of the study should be brought back into the discussion to highlight the contribution of the work.
3- Consider the sentence "...success probability of transmission, which is computed as the ratio of the number of successfully transmitted packages over the total number of packages". Why "probability"? Is the variability of this rate high?

Specific comments:
1. e.g. and i.e. -> add a comma to all occurrences - e.g., / i.e.,
2. "..interfere less with alarm messages (rarer) at reception alarm messages (rarer).." -> ??
3. "...a limited number retransmissions..." -> a limited number OF retransmissions
4. Figure 2 is not called in the text (only 2c). It should be explained, including axes.
5. "...dividing it..." -> it what? throughput?
6. "Already in Fig. 4 is presented..." -> it is presented..
7. "..there still a strong scenario-dependence.." -> there is
8. "This would required.." -> require
...
Please, revise the whole manuscript looking for typos and grammar errors.

Round 2

Reviewer 1 Report

Authors have addressed all my comments, or refuted them in a reasonable way. I have no additional remarks.

Reviewer 2 Report

The authors' review of the paper is very superficial. The answers to my comments are close to impolite. 

Point 2) of my comment has been basically disregarded. LoRaWAN (this in the title of the paper) does not allow coding rate else than 4/5 and 4/6. So either the authors clearly say that they are dealing with a proprietary system (unique and invented by them) or they must not consider other coding rates. The fact that the Semtech chip support other coding rates is immaterial for LoRaWAN. So saying "the only change we made in the manuscript is to cite this semtech's datasheet." means disregarding the comment in an impolite way.

The answer to point 9) is also basically disagreeing with my comments. To any person knowledgeable in the field the SF reservation strategy is apparently not working and no one would even try to analyze it and wasting time on something clearly unfeasible. The inclusion of this strategy seems only needed to add pages in the paper, whose findings could be summarized in a letter instead of a full paper.

Regarding point 10) I think the justification is thin air. The scientific community of telecommunications is different from that of the actuarial and economical disciplines. Anybody in the relevant audience for the journal would consider factory and agriculture as different, even in the common sense. So, one again, my comment has be purposely neglected.

Regarding point 11) the comment has not been considered. The problem of 100% is related to the calculation of the interval of confidence. That subject is covered in basic undergraduate courses. If the authors need a reference they can read "Performance Evaluation of Computer and Communication Systems" by Jean-Yves Le Boudec.

Reviewer 3 Report

All my questions have been answered and some concerns have been treated. I congratulate the authors for the work.
